# Affective and social pain modulation in children–Experimental evidence using picture viewing

**Katrin Hillmer, Judith Kappesser🆔, Christiane Hermann***

Department of Clinical Psychology, Justus-Liebig-University Giessen, Giessen, Germany

* Christiane.Hermann@psychol.uni-giessen.de

## Abstract

### Background

Children frequently encounter pain. Their pain like adults' pain is probably modulated by social-affective factors. Despite its clinical relevance, such pain modulation has not been explored experimentally in children, and little is known about specific factors accounting for it such as catastrophizing. We examined pain modulating effects of pictures varying in social-affective content and personal meaning (e.g., mothers' vs. strangers' faces) using subjective and psychophysiological measures (skin conductance, heart rate, corrugator electromyography) as outcomes.

### Methods

Forty-two children (8–13 years) underwent tonic heat pain stimulation while viewing pictures (social-affective: their mothers' faces with neutral expression, strangers' neutral and happy faces; affective: positive and negative scenes). Furthermore, the contribution of children's characteristics (e.g., anxiety, catastrophizing) and facets of the parent-child relationship to pain modulation was determined.

### Results

Viewing mothers' faces or positive scenes reduced subjective pain intensity and corrugator activity in response to pain. Viewing happy strangers' faces lowered corrugator activity. Enhanced pain experience due to negative affective stimuli was primarily observed psycho-physiologically. The correlation between children's tendency to catastrophize and pain relief by mothers' faces was mediated by induced arousal, likely reflecting the degree of motivational activation of seeking social support.

### Conclusions

Pain relief by positive affective and social-affective stimuli extends previous findings in adults, especially regarding reduced pain-related facial muscle activity. Moreover, the results shed light on the interplay between catastrophizing and social context on children's

**Data Availability Statement:** The data which were collected in this experiment are not freely and directly available because the original approval by the local ethics committee of the Faculty of Psychology and Sports Science, Justus-Liebig

University Giessen, Germany (#2010-0018) and the informed consent form signed by the participants did not include such direct access. However, the anonymized data will be made available to interested researchers upon request by the local ethics committee of the FB06 -Psychology and Sports Science. For this purpose, its contact email (ethikkommission@fb06.uni-giessen.de) is provided in the manuscript.

**Funding:** The author(s) received no specific funding for this work.

**Competing interests:** None of the authors has any competing interest.

pain experience. Clinically, our results imply that just looking at pictures of their mothers (or positive scenes) might help to alleviate pain in children.

## Introduction

Social and affective factors are both assumed to modulate pain experiences as proposed, for example, by the Socio-Communication Model of Pain [1, 2]. Social pain modulation in children has primarily been explored in experimental and clinical settings by varying parents' presence or absence [3–7]. A recent systematic review showed an overall decrease in children's pain intensity when their parents were present rather than absent during medical procedures [8]. Yet, experimentally better controlled social stimuli have not been used although this would allow to gain a better understanding of the putative underlying mechanisms on the impact of social stimuli on children's pain experience.

In adults, affective pain modulation has frequently been tested experimentally using pictures to induce affective states. In general, positive affective states reduce the pain experience, while negative affective states increase it [9–12]. To our knowledge, affective pain modulation has not been addressed experimentally in children.

Social and affective pain modulation are not independent from each other. Significant others or strangers can induce different affective states, for example, by having an acquired affective quality, or by facially expressing affective states [13–15]. For example, Hillmer and colleagues [16] found pain modulating effects in adults not only when viewing partner pictures with neutral facial expressions (but having a positive personal value), but also when viewing stranger pictures with happy facial expressions. Similarly, social-affective pain modulation in children may also vary depending on the relationship with significant others and their behavior [e.g., parents' social support or distressing behaviors; [17]. In addition, social-affective pain modulation in children may also be influenced by psychological health (e.g., anxiety, depressive symptoms), or pain-related cognitions (e.g., catastrophizing) presumably having a social-communicative function (e.g., 6, 7).

It is of interest whether pain modulation occurs not only on the level of subjective pain report but also on the level of psychophysiological measures. Pain as highly relevant stimulus induces autonomic arousal as indicated by higher skin conductance level (SCL) [18, 19], heart rate (HR) [18, 19], and higher activity of the corrugator muscle (assessed by electromyography; EMG) [20], the latter one corresponding to eyebrow squeezing as core element of the facial pain expression [21] and correlating with perceived aversiveness of stimuli [22]. A recent study with adults found effects of corrugator EMG, SCL and HR on social and affective pain modulation [16].

The present study aimed to investigate the pain modulating effect of pictures varying in affective quality (positive and negative scenes, happy and neutral faces) and social content (mothers', strangers' faces) in children during tonic heat pain stimuli. Pain modulation was assessed by self-report and physiological measures assessing autonomic arousal [(SCL, HR, and corrugator EMG; [18, 19], and pain-related facial responses, more specifically corrugator muscle activity, a core element of the facial expression of pain [21, 22]. We hypothesized that pictures of (a) negative scenes enhance, and positive scenes relieve pain, (b) neutral familiar (mother) and positive unfamiliar faces (happy stranger) alleviate pain. Finally, the influence of parental and child characteristics on children's pain modulation was tested.

## Materials and methods

### Participants

A school sample of children between the age of 8 and 14 years and their mothers were recruited via leaflets distributed at local schools between August 19[th], 2011, and December 21[st], 2012. Interested children and parents were informed about the study protocol. A telephone interview was conducted with the children's mother to determine whether the child was eligible for study participation.

Exclusion criteria were: (a) insufficient German reading, writing or speaking skills, (b) developmental delay, (c) diagnosis of mental disorder as assessed by the parent version of the Diagnostic Interview for Mental Disorders in Children and Adolescents [23], (d) uncorrected vision acuity or hearing deficits.

N = 45 children participated. Three of them had to be excluded from further analyses. One child had an extremely low pain threshold and experimental temperature (33˚C/34˚C), suggesting that the child had not properly understood the experimental instructions. One child failed to look at the monitor during picture presentation, and another one was hyperoptic, but had failed to bring her glasses. The final sample consisted of 42 children (45.2% female) between 8 and 13 years since no 14-year-old children were interested in taking part. Data of the final sample can be requested by writing to ethikkommission@fb06.uni-giessen.de.

To be able to better describe our sample of children recruited in local schools we included a question on recurrent pain. 15 of the 42 children reported heterogenous problems with recurrent pain [abdominal pain (n = 13) headaches (n = 7)] which is a considerable number. Therefore, in an attempt to control for a possible confounding influence of recurrent pain, we decided to check for statistically significant differences between both groups and to only include children with recurrent pain if they do not differ systematically from healthy children in relevant outcomes.

Demographic characteristics of the sample are displayed in Table 1.

For their participation children received 30 €. The study protocol had been approved by the local ethics committee ("Lokale Ethikkommission des Fachbereiches 06 Psychologie und Sportwissenschaften der Justus-Liebig Universität", LEK-FB06; #2010–0018; https://www.uni-giessen.de/de/fbz/fb06/psychologie/ethikkommission) stating "Decision: There are no ethical

**Table 1. Demographic characteristics.**

|  | Total sample (*n* = 42) | Healthy children (*n* = 27) | Children with recurrent pain (*n* = 15) | Comparison healthy vs. recurrent pain[c,d] |
|---|---|---|---|---|
|  | N (%) | N (%) | N (%) | $\chi^2$, *p* |
| Children's sex (female) | 19 (45.2) | 12 (44.4) | 7 (46.7) | 0.31, .580 |
| Mothers' history of chronic pain (yes) | 19 (45.2) | 9 (33.3) | 10 (66.7) | **4.33, .038** |
| Mothers' years of formal schooling[a] (9–10 years of formal schooling) | 14 (33.4) | 7 (25.9) | 7 (46.7) | 1.87, .172 |
|  | Mean (SD) | Mean (SD) | Mean (SD) | *t, p* |
| Age child[b] | 10.7 (1.7) | 10.9 (1.7) | 10.4 (1.6) | 0.84, .409 |
| Age mother[b] | 41.5 (6.2) | 41.9 (6.5) | 40.8 (5.9) | 0.56, .582 |

Demographic characteristics for the total sample as well as for healthy children and children with recurrent pain separately.

[a] 9–10 years vs. more than 10 years of formal schooling

[b] in years

[c] $\chi^2$ (df) = 1

[d] *t* (df) = 40

Note: Significant differences are highlighted in bold

or profession law objections to the project". Written informed consent was provided by all participating children and their parents.

## Questionnaires

Questionnaires were administered online via *Unipark* (Questback GmbH, Oslo, Norway) before the experimental session. Children who did not manage to fill in all required questionnaires before the experiment completed the remaining questionnaires after the experiment.

**Pain related cognitions and behavior.** *Children's pain catastrophizing* (child report) was assessed using the *Catastrophizing* subscale of the *Pain Related Cognitions Questionnaire for Children* (PRCQ-C; [24], a German and abbreviated version of the PCQ; [25]. The scale consists of 5 items which can be answered on a five-point Likert Scale, ranging from 1 (*never)* to 5 (*very often*) and start with "*When I'm in pain, I. . .*". Examples for the *Catastrophizing* scale are ". . .*worry that I will always be in pain*", or ". . . *think that nothing helps*". The scale shows satisfactory internal consistency (*Cronbach's α* = .78) and a high test-retest reliability *(r_{tt}* = .86; 24). Further, good construct validity has been demonstrated [24].

*Mothers' solicitous response to pain* (child report) was assessed using the children's version of the *Solicitous Response* scale of the *Pain-related Parent Behavior Inventory* (PPBI-C; 26). The scale consists of 6 items which are to be answered on a 5-point Likert scale ranging from 1 (*never*) to 5 (*very often*) and start with "*When I'm in pain, my mother. . .*" Examples are ". . .*takes special care of me.*", ". . .*takes over my chores and duties.*". Internal consistency is good (*Cronbach's α* = .81) as is its construct validity [26].

*Mothers' catastrophizing about her child's pain* (mother report) was assessed by the German version of the *Pain Catastrophizing Scale-Parents* [PCS-P [27]; German version [28]. The PCS-P consists of 13 items which describe thoughts and feelings that parents may experience when their child is in pain. All items start with "When my child is in pain. . ." and are assigned to one of three subscales: *Helplessness (e.g., . . ."I cannot do anything to stop the pain"*), *Rumination (e.g., . . ."I'm worrying all the time, if the pain will end any time"*), and *Magnification* (e.g., . . ."I'm afraid, that the pain indicates something bad"*). The items are to be answered on a 5-point Likert scale ranging from 0 (*not at all*) to 4 (*extremely*). The PCS-P has high internal consistency ☻*Cronbach's α* = .93; [27, 28] and good validity [27].

**Measures of children's psychological health.** *Depressive symptoms* (child report) was assessed using the *Depression Test for Children* (DTC; 29). This self-report screening instrument contains dichotomous 55 items. The DTC entails three subscales: *Dysphoric Mood and Self-esteem Problems* (e.g., "Are you often so sad that you can hardly stand it?", "Do you often think that you are a bad person?"), *Agitated Behavior* (e.g., "Are you often warned to be quiet at school?"), and *Fatigue and Psychosomatic Symptoms* (e.g., "Are you often tired all day long?", "Do you have more pain than other children?"). Each of the three subscales has good internal consistency (*Cronbach's α* = .86/.78/.75) and retest reliability [r_{tt} = .88/.89/.82); [29]. Factorial and content validity have been established [29].

*Anxiety* (child report) was assessed using Form A of the *Children Anxiety Test II* (CAT-II; [30]. The 20 yes/no items assess dispositional anxiety. Examples are "*I'm worrying about the future.*", "*I often feel anxious.*". The authors report good internal consistency (*Cronbach's α* = .80) and retest reliability (*r_{tt}* = .80) and several studies showing satisfactory construct validity [30].

*Emotional and behavior problems* (mother report) were assessed using the problem score of the parent version of the *Strength and Difficulties Questionnaire* [SDQ: [31]; German version: [32]. The questionnaire consists of five scales with 25 items in total: *Emotional Problems*, *Conduct Problems*, *Hyperactivity/Inattention*, *Peer Problems* and *Prosocial Behavior*. The items are

to be answered on a 3-point-Likert scale: 0 (*not true*), 1 (*somewhat true*), 2 (*certainly true*). The problem score is determined by adding the scores of the four problem-related subscales. The questionnaire has good reliability and validity in healthy and clinical samples of 5- to 17-year-old children and adolescents [33, 34]. The factor structure and construct validity of the subscales could be replicated for the German version [32, 35]. The problem score showed good internal consistency [*Cronbach's α* = .80/.83; [32, 35].

**Parenting style.** *Mothers' Support and Approval* (child report) was assessed using the *Childrearing Style Inventory* ☉CSI: [36]. **The self-report questionnaire assesses potentially problematic child-rearing behaviors of parents. It consists of five different subscales, two of which were used here measuring** *Support* **and** *Approval*. Each of the two subscales entails 12 items which can be answered on a 4-point Likert scale ranging from 1 *(never or very seldom)* to 4 (*always or almost always)*. Examples for the *Support* scale are: "My mother shows me how things work that I want to deal with." or "When an important decision has to be made (e.g. vacation trip), my mother also listens to my opinion.". Examples for the *Approval* scale are: "My mother is happy when I helped with a job." or "My mother is happy when I share something with my siblings or friends.". Good internal consistency (*Kuder Richardson;* support $r_{it}$ = .78-.90, approval $r_{it}$ = .89-.92), retest-reliability (support $r_{tt}$ = .57-.72, approval $r_{tt}$ = .64-.69) and construct validity of the scales are reported [36, 37].

## Experimental set-up

The experimental design corresponds to the experimental design used in a previous study with female adults [16]. The experiment consisted of 15 trials of tonic heat pain stimulation with simultaneous picture presentation (for a detailed overview see Figs 1 and 2). The inter-trial-interval varied randomly between 15 and 35 s.

**Heat pain stimulation.** All heat stimuli were applied with a 30 x 30 mm Peltier element-based advanced thermal stimulator (ATS) thermode (Pathway Model Cheps, Medoc Ltd, 2005, Ramat Yishai, Israel) placed on the thenar of the non-dominant hand.

In each experimental trial, tonic heat stimulation started from a baseline temperature of 32˚C, with temperature increasing at a rate of 1˚C/s until the individually adjusted target temperature was reached and maintained for 54 s. Then, temperature returned to baseline at a rate of 8˚C/s.

The intensity of the experimental tonic heat pain stimuli was individually adjusted such that (a) the intensity of the tonic heat pain stimulus was rated as about 10 on a visual analogue scale (VAS, 0–20 units), and that (b) pain intensity would not be rated below 8 on the VAS during the 50 s adjustment trial stimulation. To determine the individual stimulus intensity, a two-step approach was followed: (a) The heat pain threshold (HPT) was determined using the method of limits. Following three practice trials, five HPT trials were run. HPT was defined as the mean of the last three trials; (b) The temperature for the tonic heat pain stimulation was determined by a series of tonic heat trials (adjustment trials) with an increase in stimulation temperature. Each trial during the adjustment phase lasted 50 s, started at a baseline temperature of 32˚C which increased at a rate of 1˚C/s until reaching the target temperature. In the first trial, the target temperature was set at 1˚C below HPT. In each subsequent trial, the temperature of the tonic heat stimulus was increased by 0.5˚C. For safety reasons the maximum temperature was set to 49˚C. During the 50 s tonic stimulation, the participants rated the pain intensity continuously using the VAS. Subsequently, the temperature returned to baseline at a rate of 8˚C/s. After an inter-trial-interval of 30 s, participants started the next heat stimulus by pressing the Enter button. The calibration procedure was terminated when pain intensity was rated at about 10 and not less than 8 on the VAS units.

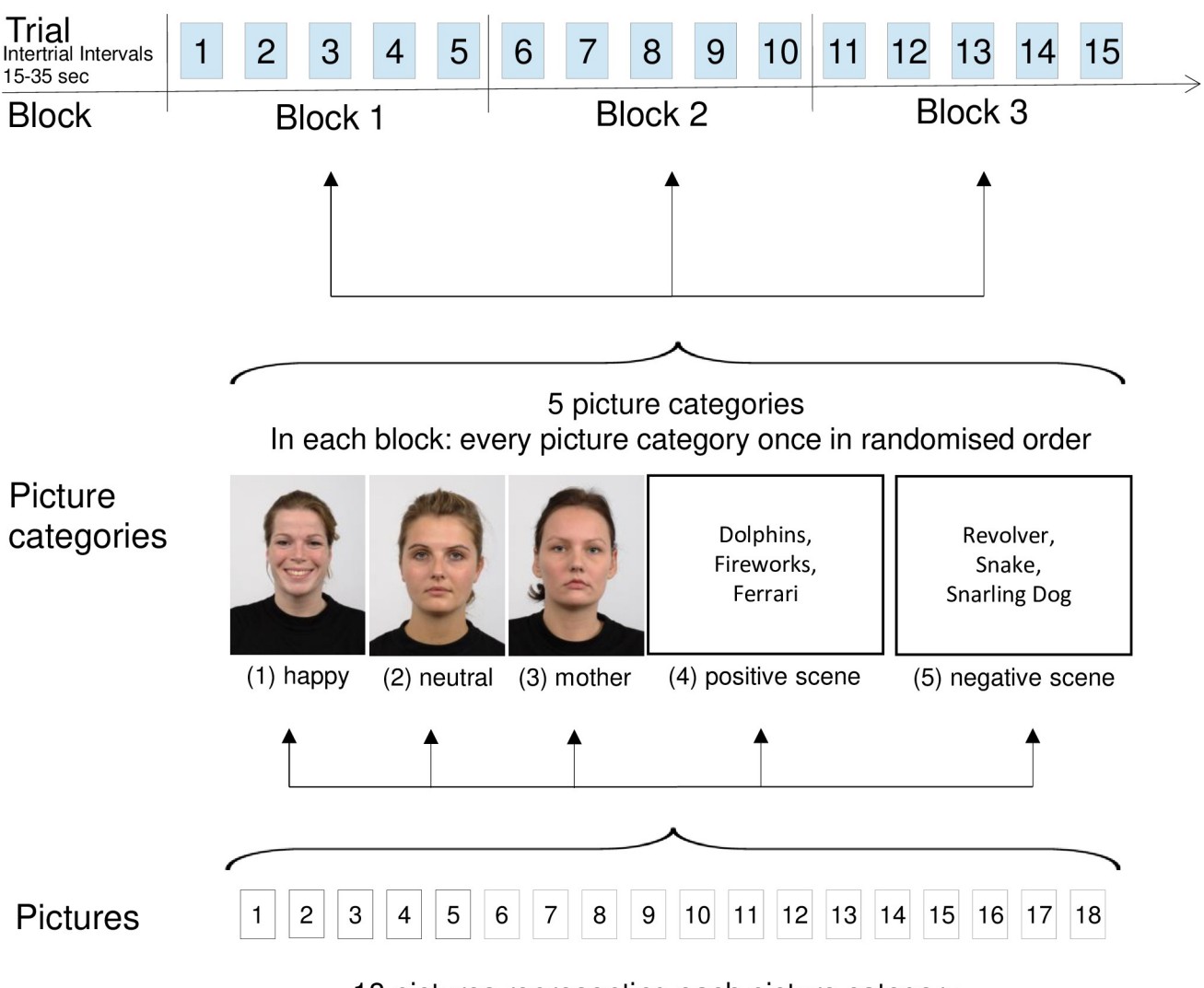

**Fig 1. Experimental design.** Each picture category was presented once in each of the three blocks (block 1–3). In each block, the order of the five picture categories was randomized for every participant. For each trial six pictures from the set of 18 pictures of one category were pseudo-randomly chosen so that every picture was shown once across all trials. Inter-trial-intervals (ITI) varied between 15 and 35 s. Note: Image codes of the exemplarily displayed pictures for each picture category from left to right: Rafd090_02 (happy), Rafd090_31(neutral), Rafd090_08 (as example for mother due to picture rights), no examples for IAPS pictures could be given due to unclear copyright situation.

**Picture stimuli.** Five categories of pictures were presented to investigate social-affective and affective pain modulation: (a) participants children's mothers' neutral faces (from now on referred to as 'dmothers' faces'), (b) standard strangers' neutral faces ('neutral faces'), (c) standard strangers' happy faces ('happy faces'), (d) positive scenes, and (e) negative scenes. In order to be able to separate the effect of an expressed emotion ('happy') and the personal meaning of the social cue ('mother'), a neutral facial expression of the mother was used.

For every picture category a set of 18 pictures was chosen. Happy and neutral faces were taken from the Radboud Face Database [38] and the Karolinska Directed Emotional Faces picture set [39]. Faces were chosen on the basis of the best emotion recognition hit rates and the

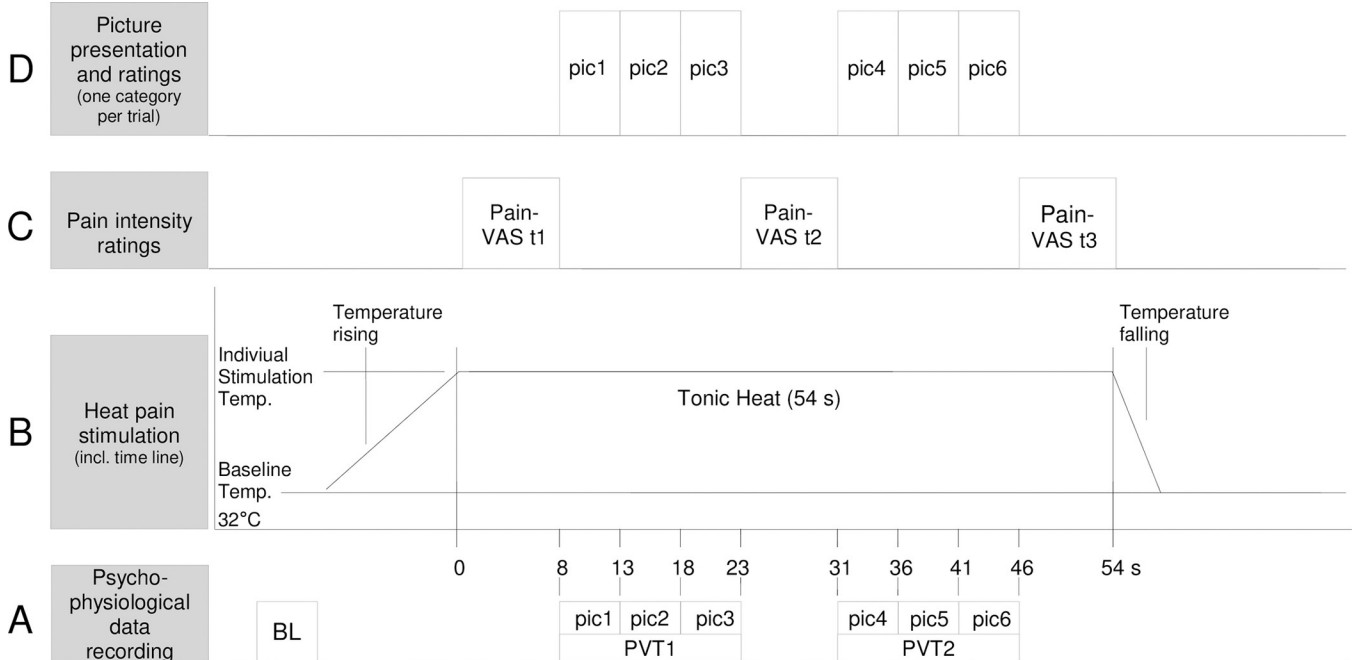

**Fig 2. Trial structure.** The figure shows the time course of one experimental trial, i.e. for six pictures of one picture category. (A) Psychophysiological data were recorded continuously and later segmented [baseline (BL): 5 s before temperature rising; picture viewing time 1 (PVT1): the three 5 s intervals for presentation of pic1 to pic3; PVT2: the three 5 s intervals for presentation of pic4 to pic6]. (B) The heat pain stimulation started from a baseline temperature of 32°C and increased at a rate of 1°C/s until reaching the individually adjusted stimulation temperature. The target temperature was held constant for 54 s and then returned to baseline temperature at a rate of 8°C/s. (C) Pain intensity was rated on a visual analogue scale (VAS). The VAS were shown on the screen for 8 s and participants were asked to rate pain intensity once prior to picture viewing (Pain-VAS t1) and twice after each picture viewing time (Pain-VAS t2: after PVT1; Pain-VAS t3: after PVT2). (D) Picture presentation with ITIs between 15 and 35 s.

highest emotion intention scores [38, 39]; Codes used: happy: AF34HAS, AF06HAS, AF07HAS, AF11HAS, AF20HAS, AF22HAS, AF25HAS, AF26HAS, Rafd090_02, Rafd090_12, Rafd090_14, Rafd090_16, Rafd090_19, Rafd090_22, Rafd090_26, Rafd090_27, Rafd090_37, Rafd090_56; neutral: AF01NES, AF03NES, AF06NES, AF07NES, AF13NES, AF19NES, AF26NES, AF29NES, AF34NES, Rafd090_56, Rafd090_57, Rafd090_08, Rafd090_12, Rafd090_61, Rafd090_26, Rafd090_31, Rafd090_32, Rafd090_37]. Only female faces were chosen to keep the sex of faces pictures constant. The 562x762 pixels (Karolinska Directed Emotional Faces) or 572x762 pixels (Radboud Face Database) pictures were colored and in portrait format. Examples are shown in Fig 1.

Based on valence and arousal ratings, all negative and some positive scenes were taken from the International Affective Picture System [IAPS; [40]; IAPS codes used: negative: 1120, 1201, 1300, 1525, 1930, 1932, 3500, 6230, 6243, 6250, 6312, 6313, 6315, 6350, 6510, 6520, 6550, 6560; positive: 1710, 1722, 1811, 1920, 8490, 8496]. As many pleasant IAPS pictures are not suitable for children (e.g. erotic ones), we searched the internet for more appropriate pleasant pictures such as dolphins or leisure park scenes. In a pilot study, we asked a sample of $n = 287$ children aged 8 to 14 to rate valence and arousal of these pictures. Based on the best valence and arousal ratings, we chose six IAPS pictures and 12 from our own picture set. Valence and arousal ratings for the 12 pictures from our set can be found in S1 Table. These pictures were colored, in landscape format and their pixel size varied between 1000 x 800 and 1024 x 654–769.

The frontal pictures ($n = 18$) of participating children's mothers were either taken at home before the day of the experiment or in the laboratory on the day of the experiment under

standardized conditions. Mothers were instructed to wear a grey, black, or white T-shirt, take the photos in absence of the children, and look straight into the camera without smiling or expressing any other kind of emotion. All pictures were checked for deviations from a neutral expression by the two experimenters, one of which was always one author (K.H.). Whenever pictures were not considered to be neutral, new pictures were taken in the laboratory on the day of the experiment. The final pictures were matched in size (561x761 pixels) to the standardized faces.

There were three trials per picture category presented in a pseudo-random order so that the same picture category was not shown more than twice in consecutive trials (see Fig 1). The 18 pictures in each picture category were pseudo-randomly assigned to one of the three trials so that every picture was shown only once during the trials.

**Trial structure.**   Experimental design and trial structure are illustrated in Figs 1 and 2. Each experimental trial consisted of a tonic heat stimulus with a duration of 54 s. During each tonic heat stimulus six pictures of one of the five picture categories were presented, each picture for 5 s. Presentation of the pictures 1–3 is further referred to as picture viewing time 1 (PVT1; total duration: 15 s), the presentation of the pictures 4–6 as PVT2 (total duration: 15 s).

**Subjective outcome measures.**   *Pain intensity* was rated three times per trial on a visual analogue scale (VAS) presented on the PC monitor ranging from no pain (0) to worst pain imaginable (20) within a time window of 8 s: (1) VAS t1—immediately when the target experimental temperature was reached and prior to picture viewing, (2) VAS t2 –after the end of PVT1, and (3) VAS t3 –after PVT2 (see Fig 2).

*Valence and arousal.* Separate picture rating trials to rate valence and arousal of the pictures (see Fig 3) were included after the experimental trials to avoid excessive demand by additional ratings during the experimental trials. In each picture rating trial six pictures of one picture category were presented in random order. There were three trials per picture category and 15 picture ratings trials altogether. The three trials of one picture category were presented in a pseudo-random order, i.e. the same picture category was not shown more than twice in consecutive trials, and all 18 pictures of one picture category were pseudo-randomly assigned to one of the three trials so that every picture was shown only once during the three trials of one category. Children were asked for one valence and one arousal rating per trial averaged over the six pictures presented (see Fig 3).

Valence and arousal ratings were obtained using computerized versions of the well-established Likert self-assessment manikins [SAM; [41]; valence: 1 = very unpleasant; 9 = very pleasant; arousal: 1 = very calm; 9 = very arousing]. Children were trained how to use the SAM before the start of the experimental trials.

**Psychophysiological outcome measures.**   All psychophysiological signals were recorded continuously with a QuickAmp 72 Amplifier using BrainVision Recorder software (Brain Products GmbH, 2008) and sampled at 1000 Hz. A ground electrode was placed on the right clavicle. Children were instructed not to move their hands, except when asked to rate. Markers were recorded such that the data segments for BL, PVT1 and PVT2 could be extracted offline. All other recordings over time were not considered for analysis.

*Skin conductance level (SCL).* For SCL recording, a galvanic skin response adaptor (GSR sensor, Brain Products GmbH) and two Ag/AgCl Electrodes (diameter: 10 mm) filled with TD-246 isotonic paste (EASYCAP GmbH, Herrsching, Germany) were used. The electrodes were placed on the thenar eminence of the dominant hand after having cleansed the hand with water.

*Corrugator electromyography (Corrugator EMG).* Activity of the M. corrugator superciliisupercilii was measured using a bipolar reording with two Ag/AgCl electrodes (diameter: 5mm) filled with electrolyte-gel (SuperVisc, Easycap GmbH). After cleansing the skin with alcohol,

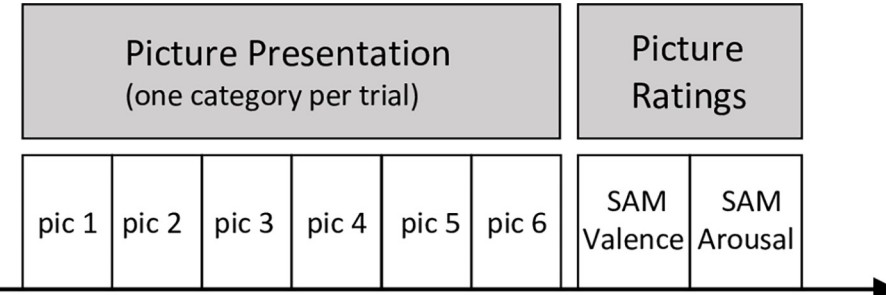

**Fig 3. Trial structure of picture rating trials.** In every rating trial six pictures of one category were presented for 5 s, followed by one valence and one arousal rating per trial. Valence and arousal were rated using the self-assessment manikin (SAM). Randomization procedure for the pictures corresponds to that of experimental trials (see Figs 1 and 2).

the electrodes were placed above the left eyebrow according to EMG recording guidelines (Fridlund and Cacioppo 1986).

*Heart rate (HR).* Heart rated was measured using an electrocardiogram (ECG) with two pre-gelled AG/AgCl electrodes (Megro, Wesel, Germany). One electrode was placed on the right clavicle and the other one on the lowest left rib.

## Procedure

When a child was considered eligible, its mother was sent an email including a standardized instruction on how to take the necessary neutral portrait photos. Most mothers decided to be photographed in the laboratory, however. All mothers provided informed consent for the use of their photos in the study.

The experiment took place in a psychophysiology lab at the local university. The experimental chamber was equipped with a video camera, a microphone, and speakers to allow communication with and monitoring of the children from an adjacent control room. Experimental control (visual stimuli, ratings, trigger for physiological data recording and heat pain stimulation) was realized with Presentation software (Neurobehavioral systems, Inc, Version 14.2) from the adjacent control room.

When mothers and children arrived at the university, they were shown and explained the experimental chamber, the room in which they completed the questionnaires on PC, and the waiting area. Then they were familiarized with the heat pain device and provided informed consent. Next, the children were accompanied by a research assistant to a room in which they started filling in the questionnaires. Meanwhile the mothers remained in the experimental room where photos of them were taken and prepared for the use in the experiment. If mothers had already provided photos, they were asked to wait for 20 minutes in the waiting area. After the photos had been taken, the mothers were accompanied to the room in which the children were filling in the questionnaires. Now mothers were placed in front of a computer to fill in questionnaires and their children were picked up to participate in the experiment. After having filled in all questionnaires, mothers were asked to take a seat in the waiting area until their children had finished the experiment.

For the experiment the children were seated comfortably in a chair with armrests. On a desk in front of the children were a PC with a 20" monitor (resolution: 1680 x 1050 pixels, distance = 0.5 m) for picture and rating scale presentation, a mouse and a computer keyboard. Next, the thermode was attached to the non-dominant hand of the children with a self-adhesive fixation bandage. Heat pain thresholds were determined, and the experimental

temperature was adjusted. After the sensors for physiological recording had been applied, the 15 experimental trials started. The actual experiment lasted about 20 min. After the trials the thermode was removed, and the children rated pictures' valence and arousal in the picture rating trials for valence and arousal ratings. After these trials all electrodes were removed.

Children who had not managed to finish filling in all required questionnaires before the experiment went back to the room with the PC in order to complete them. Finally, all children and mothers were debriefed, had the opportunity to ask questions and received their payment.

## Data preprocessing

**Ratings.** The mean pain intensity was calculated as the average over three trials for each picture category and each rating time (VAS t1, VAS t2, VAS t3). Mean valence and arousal for each picture category were also calculated by averaging the ratings over the three trials per category.

**Calculation of pain modulation indices.** For correlational analyses we defined three different modulation indices, reflecting positive affective, negative affective, and social pain modulation. We decided to calculate difference scores between positive scenes, negative scenes, mothers' and happy faces on one hand and trials with neutral faces on the other hand to account for individual differences in pain experience. (1) *Positive pain modulation (posPM)* was defined as the difference in pain intensity between trials with neutral faces and positive scenes (neutral faces–positive scenes) for VAS t2 and t3, respectively. Accordingly, PosPM scores larger than 0 describe a pain-relieving effect of positive scenes compared to neutral faces. (2) *Negative pain modulation (negPM)* was defined as the difference in pain intensity between negative scenes and neutral faces (negative scenes–neutral faces) for VAS t2 and t3, respectively. Accordingly, NegPM scores larger than 0 describe a pain-increasing effect of negative scenes compared to neutral faces. (3) *Social pain modulation (motherPM, happyPM)* was defined as the difference in pain intensity between neutral faces and mothers' faces or happy faces (motherPM: neutral faces–mother faces; happyPM: neutral faces–happy faces) for VAS t2 and t3, respectively. Accordingly, motherPM and happy PM scores larger than 0 describe a pain-relieving effect of the mothers' and happy pictures compared to neutral faces.

**Psychophysiological data.** Preprocessing and aggregation of the psychophysiological data was conducted with customized MATLAB programs ☉version R2011a, MathWorks].

The SCL signal was filtered online with a 100 Hz low pass filter. The mean SCL (mS) for 500 ms epochs was calculated. All 500 ms epochs containing artefacts were excluded from further analysis.

The corrugator EMG was filtered online with a 10Hz high pass filter. The EMG signal was filtered offline with a 50 Hz notch filter and visually inspected for artefacts. The EMG signals were rectified and integrated using contour-following integrators with a time constant of 80 ms [42]. Mean EMG activity (μV) was calculated for 500 ms epochs. All 500ms epochs containing artefacts were excluded from further analysis.

ECG signals were filtered online with a 100 Hz low pass filter and a 0.5Hz high pass filter to avoid baseline shifts in the ECG [43]. The signal was visually inspected for artefacts and R-Wave detection errors. For HR analysis, beat-to-beat intervals were calculated and transformed into beats per minute for every data point and averaged across 500 ms epochs. All 500 ms epochs containing artefacts were excluded from further analysis.

To reduce inter-individual differences all physiological raw data were intraindividually z-transformed [$z_i = (x_i - M_i)/SD_i$]. As we used tonic heat pain stimulation, we were interested in

psychophysiological responses during the heat pain stimulation. Therefore, after standardization (z-scores), we computed mean values for seven 5 s intervals per trial (BL, pic1-pic6), which were further collapsed over the three trials per picture category. Finally, averages for PVT1 (i.e. mean of pic1-pic3) and PVT2 (i.e. mean of pic4-pic6) were computed, yielding mean values for PVT1, PVT2, and BL (see Fig 2).

## Statistical analysis

All statistical analyses were calculated using IBM SPSS Statistics (version IBM 20.0.0, IBM Corp., Armonk, NY, 2011). Level of significance was set to $\alpha$ = .05 (two-tailed) for all analyses.

**Analyses of variance.** We used the GLM procedure to conduct mixed design ANOVAs to analyze valence and arousal, pain intensity modulation, and psychophysiological correlates depending on picture category and time as within-subject factors. Since a number of children reported recurrent pain experiences we also included recurrent pain ("GROUP") as between-subject factor to explore potential differences. Significant interactions with GROUP were followed by separate ANOVAS for healthy children and children with recurrent pain. Yet, since there were no significant effects for GROUP in any of the GLM analyses for the subjective and the physiological measures, only the results of the GLM analyses based on the total sample are reported.

If the assumption of sphericity was violated, Greenhouse-Geisser corrected degrees of freedom (*df*) were used, yet, the nominal *df* are reported. For *F*-tests partial eta squared ($\eta^2$) was calculated as effect size with 0.099 as threshold for small, 0.0588 for medium and 0.1379 for large effect sizes according to Cohen [44, 45].

*Picture ratings*. Differences in valence and arousal ratings depending on picture categories and effects of recurrent pain were analyzed by two two-way (5x2) mixed-design ANOVAs with the within-subject factor PICTURE CATEGORY (5: negative scenes, positive scenes, neutral faces, happy faces, mothers' faces) and the between-subject factor GROUP (2: healthy children, children with recurrent pain). Significant interactions between PICTURE CATEGORY and GROUP were followed by separate ANOVAS for healthy children and children with chronic pain. In case of a significant main effect of PICTURE CATEGORY post-hoc Bonferroni adjusted estimated marginal means tests were conducted using the COMPARE function of the GLM procedure (Howell and Lacroix 2012) to analyze all pairwise contrasts between the five picture categories.

*Modulation of pain intensity and psychophysiological reactions*. The pain modulation by picture category, time and effects of recurrent pain was analyzed using a three-way (5x3x2) mixed-design ANOVA with the within-subject factors PICTURE CATEGORY (5: negative scenes, positive scenes, neutral faces, happy faces, mothers' faces) and TIME (3: VAS t1, VAS t2, VAS t3) and the between-subject factor GROUP (2: healthy children, children with recurrent pain).

The psychophysiological correlates (corrugator EMG, SCL, HR) were analyzed using three-way (5x3x2) mixed design ANOVAs with the factors PICTURE CATEGORY (5: negative scenes, positive scenes, neutral faces, happy faces, mothers' faces) and TIME (3: BL, PVT1, PVT2), and the between-subject factor GROUP (2: healthy children, children with recurrent pain).

Significant interactions between TIME and PICTURE CATEGORY were decomposed by conducting separate ANOVAS for every time point with the factor PICTURE CATEGORY.

Modulation was defined as contrast to the neutral faces category. Therefore, in case of main effects of PICTURE CATEGORY at a certain time point (pain intensity: t1-t3; corrugator EMG, SCL, HR: BS, PVT1, PVT2) planned simple contrasts were conducted comparing the

social and affective picture categories with neutral faces. As effect size for mean comparisons, Cohen's *d* for repeated measures was used [$d_{rm}$; [46]]. To control Type I error rate inflation by multiple contrasts, the significance level for simple contrasts was divided by the number of contrasts ($\alpha$ = .05/4 = .0125).

**Correlational analyses.** We used Pearson's product-moment correlation coefficients to determine the relationship between the pain modulation indices (posPM, negPM, motherPM, happyPM), selected questionnaires tapping child characteristics (PRCQ-C, DTC, CAT-II, SDQ) and valence and arousal ratings for the corresponding picture category. MotherPM was additionally also correlated with mothers' pain-related catastrophizing (PCS-P), pain-related behaviors (PPBI-C) and parenting style (CSI). If associations hinted at a mediation, i.e. one of the questionnaires correlated significantly with another questionnaire and the modulation indices, partial correlations were calculated to determine the unique relationship between the variable and the modulation index.

## Results

### Questionnaires

Children with recurrent pain differed from healthy children in measures of psychological health. Mothers of children with recurrent pain reported higher depressed mood and anxiety in their children and more difficulties faced by their children than mothers of healthy children. Children with recurrent pain reported to get more approval from their mothers than healthy children do. No differences were found between both groups of children with regard to parenting style, pain catastrophizing or parental pain behaviors (see Table 2 for all results).

### Valence and arousal ratings for each picture category

Since there were no significant effects for the between-subject factor GROUP in any of the GLM analyzes, we report results of all GLM analyses based on the total sample here.

Analysis of the valence and arousal ratings found a significant main effect of PICTURE CATEGORY ($F(4,164)$ = 45.97, $p$ < .000, $\eta^2$ = 0.53; $F(4,164)$ = 73.88, $p$ < .000, $\eta^2$ = 0.64; see Table 3).

With regard to valence, positive scenes were perceived as most pleasant, followed by mother and happy faces which evoked more pleasure than neutral faces. Negative scenes were perceived to be most unpleasant. With regard to arousal, negative scenes were perceived to be most arousing, followed by positive scenes. Mother and happy faces did not differ in the related subjective arousal, but mother faces differed from neutral faces which were perceived as least arousing.

### Modulation of pain intensity by pictures over time

The 5x3 repeated measure ANOVA of pain intensity revealed significant main effects of PICTURE CATEGORY, TIME, and a significant PICTURE CATEGORY x TIME interaction effect ($F(4,164)$ = 5.05, $p$ = .002, partial $\eta^2$ = .11; $F(2,82)$ = 4.65, $p$ = .027, $\eta^2$ = .10; $F(8,328)$ = 3.72, $p$ = .004, $\eta^2$ = .083). The interaction was decomposed into three separate ANOVAS for PICTURE CATEGORY at every time point (t1: $F(4,164)$ = 0.27, $p$ = .900, partial $\eta^2$ = .01; t2: $F(4,164)$ = 5.81, $p$ = .002, partial $\eta^2$ = .11; t3: $F(4,164)$ = 6.03, $p$ = .001, $\eta^2$ = .13; see Fig 4).

At t1, there were no differences between neutral and other pictures. At t2, more pain was reported when viewing neutral as compared to mothers' faces ($d_{rm}$ = .34) or positive scenes ($d_{rm}$ = .26). At t3, more pain was reported when viewing neutral faces as compared to positive scenes ($d_{rm}$ = .26).

**Table 2. Questionnaire reports.**

| | Total sample (n = 42) | Healthy children (n = 27) | Children with recurrent pain (n = 15) | Comparison healthy vs recurrent pain |
|---|---|---|---|---|
| | Mean (SD) | Mean (SD) | Mean (SD) | $t$ (df), $p$ |
| *Catastrophizing and parental pain behavior* | | | | |
| Child report: | | | | |
| Catastrophizing (PRCQ-C) | 1.71 (0.45) | 1.61 (0.41) | 1.89 (0.48) | -1.98 (40), .055 |
| Solicitous response (PPBI-C) | 3.85 (0.66) | 3.89 (0.57) | 3.79 (0.80) | 0.47 (40), .642 |
| Mother report: | | | | |
| Catastrophizing about the child's pain (PCS-P) | 16.43 (8.39) | 17.04 (8.13) | 15.33 (9.02) | 0.63 (40), .535 |
| *Psychological well-being* | | | | |
| Child report: | | | | |
| Depression (DTC) | 12.95 (9.22) | 9.41 (6.45) | 19.33 (9.96) | **-3.46 (40), .002** |
| Anxiety (CAT-II) | 3.62 (3.40) | 2.22 (2.41) | 6.13 (3.54) | **-4.25 (40), .000** |
| Mother report: | | | | |
| Child's difficulties (SDQ—problem score) | 11.79 (4.28) | 9.89 (3.25) | 15.20 (3.82) | **-4.77 (40), .000** |
| *Parenting style* | | | | |
| Child report: | | | | |
| Mothers' support (CSI) | 36.62 (6.08) | 35.89 (6.92) | 37.93 (4.08) | -1.20 (40), .236 |
| Mothers' approval (CSI) | 41.12 (6.74) | 38.96 (7.01) | 45.00 (4.09) | **-3.52 (40), .001** |
| *Experimental pain measures* | | | | |
| Heat pain threshold (°C) | 41.36 (3.27) | 41.47 (3.18) | 41.15 (3.55) | 0.30 (40), .763 |
| Experimental temperature (°C) | 42.10 (2.94) | 42.26 (2.79) | 41.80 (3.27) | 0.48 (40), .633 |
| VAS t1 | 8.04 (3.76) | 7.74 (3.60) | 8.57 (4.12) | -0.69 (40), .497 |

Questionnaire reports for the total sample and, separately, for healthy children and children with recurrent pain as well as the comparison between healthy children and children with recurrent pain.

Note: Significant differences are highlighted in bold.

Abbreviations: PRCQ-C = Pain Related Cognitions Questionnaire for Children; PPBI-C = Pain-related Parent Behavior Inventory–Child version; PCS-P = Pain Catastrophizing Scale–Parent version; DTC = Depression Test for Children; CAT-II = Children Anxiety Test II; SDQ = Strength and Difficulties Questionnaire; CSI = Childrearing Style Inventory; VAS: Visual analogue scale; t1 = before picture presentation.

Means and standard deviations for all pain modulation indices are listed in S2 Table in the Supplementary Information.

## Psychophysiological responses during tonic heat pain and picture viewing

Time courses of corrugator EMG, SCL, and HR are displayed in Fig 5.

**Corrugator EMG.** The analysis of corrugator EMG revealed main effects of PICTURE CATEGORY, TIME, and a PICTURE CATEGORY x TIME interaction ($F(4,164) = 9.33$, $p <$

**Table 3. Valence and arousal ratings.**

| Picture | Negative scenes | Neutral faces | Happy faces | Mother faces | Positive scenes |
|---|---|---|---|---|---|
| | *Mean (SD)* | *Mean (SD)* | *Mean (SD)* | *Mean (SD)* | *Mean (SD)* |
| **Valence** | 3.36[a] (2.31) | 5.08[b] (1.88) | 6.14[c] (2.11) | 7.07[c] (1.73) | 7.93[d] (1.09) |
| **Arousal** | 6.63[a] (2.12) | 1.76[b] (1.10) | 2.06[b,c] (1.26) | 2.66[c] (2.03) | 5.06[d] (2.52) |

Means and standard deviations (*SD*) for the five picture categories for the total sample (*n* = 42).

[abcd] Significant differences (*p* < .05) between categories are indicated by different superscripts.

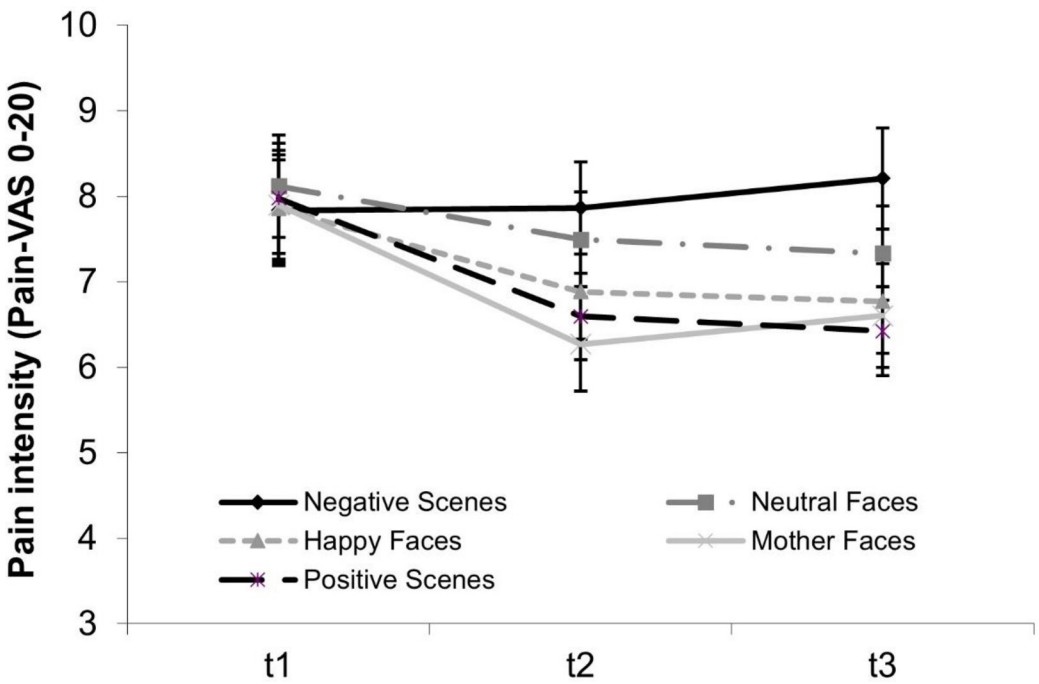

**Fig 4. Means and standard errors for pain intensity ratings.** Participants rated their pain intensity on a VAS 0–20 during tonic heat pain stimuli before picture viewing (t1) and after picture viewing (t2, t3) with t2: picture viewing time 1 (PVT1) including pictures 1–3 and t3: PVT2 including pictures 4–6.

.001, $\eta^2$ = .19; $F(2,82)$ = 11.76, $p <$ .001, $\eta^2$ = .22; $F(8,320)$ = 5.90, $p <$ .001, $\eta^2$ = .13). The interaction was decomposed into three separate ANOVAS for PICTURE CATEGORY at every time point (t1: $F(4,164)$ = 0.28, $p$ = .890, $\eta^2$ = .01; t2: $F(4,164)$ = 13.03, $p <$ .000, $\eta^2$ = .24; t3: $F(4,164)$ = 10.50, $p <$ .000, $\eta^2$ = .20; see Fig 5A).

At t1, there were no differences in corrugator activity between neutral and other pictures. At t2, corrugator activity was higher when viewing neutral faces as compared to mothers' faces, happy faces and positive scenes ($d_{\mathrm{rm}}$ = 1.42, $d_{\mathrm{rm}}$ = .54, $d_{\mathrm{rm}}$ = 1.41). At t3, corrugator activity was higher when viewing neutral faces as compared to mothers' faces, happy faces and positive scenes ($d_{\mathrm{rm}}$ = .99, $d_{\mathrm{rm}}$ = .68, $d_{\mathrm{rm}}$ = 1.08).

**Skin conductance level.** The analysis of skin conductance level (SCL) revealed a main effect of TIME, and a PICTURE CATEGORY x TIME interaction ($F(2,82)$ = 12.25, $p <$ .001, $\eta^2$ = .23; $F(8,328)$ = 6.26, $p <$ .001, $\eta^2$ = .13). The interaction was decomposed into three separate ANOVAS for PICTURE CATEGORY at every time point (t1: $F(4,164)$ = 1.06, $p$ = .377, $\eta^2$ = .03; t2: $F(4,164)$ = 3.65, $p$ = .007, $\eta^2$ = .08; t3: $F(4,164)$ = 4.94, $p$ = .002, $\eta^2$ = .11; see Fig 5B).

At t1, there were no differences in SCL between neutral and other pictures. At t2, SCL was lower for neutral faces as compared to negative scenes ($d_{\mathrm{rm}}$ = 0.76). At t3, SCL was lower for neutral faces as compared to negative scenes ($d_{\mathrm{rm}}$ = 0.92).

**Heart rate.** The analysis of heart rate (HR) revealed main effects of PICTURE CATEGORY, TIME, and a PICTURE CATEGORY x TIME interaction ($F(4,164)$ = 8.31, $p <$ .001, $\eta^2$ = .17; $F(2,82)$ = 14.37, $p <$ .001, $\eta^2$ = .26; $F(8,328)$ = 2.83, $p$ = .013, $\eta^2$ = .07). The interaction was decomposed into three separate ANOVAS for PICTURE CATEGORY at every time point (t1: $F(4,164)$ = 2.46, $p$ = .048, $\eta^2$ = .06; $F(4,164)$ = 5.88, $p$ = .001, $\eta^2$ = .13; $F(4,164)$ = 10.25, $p <$ .001, $\eta^2$ = .20; see Fig 5C).

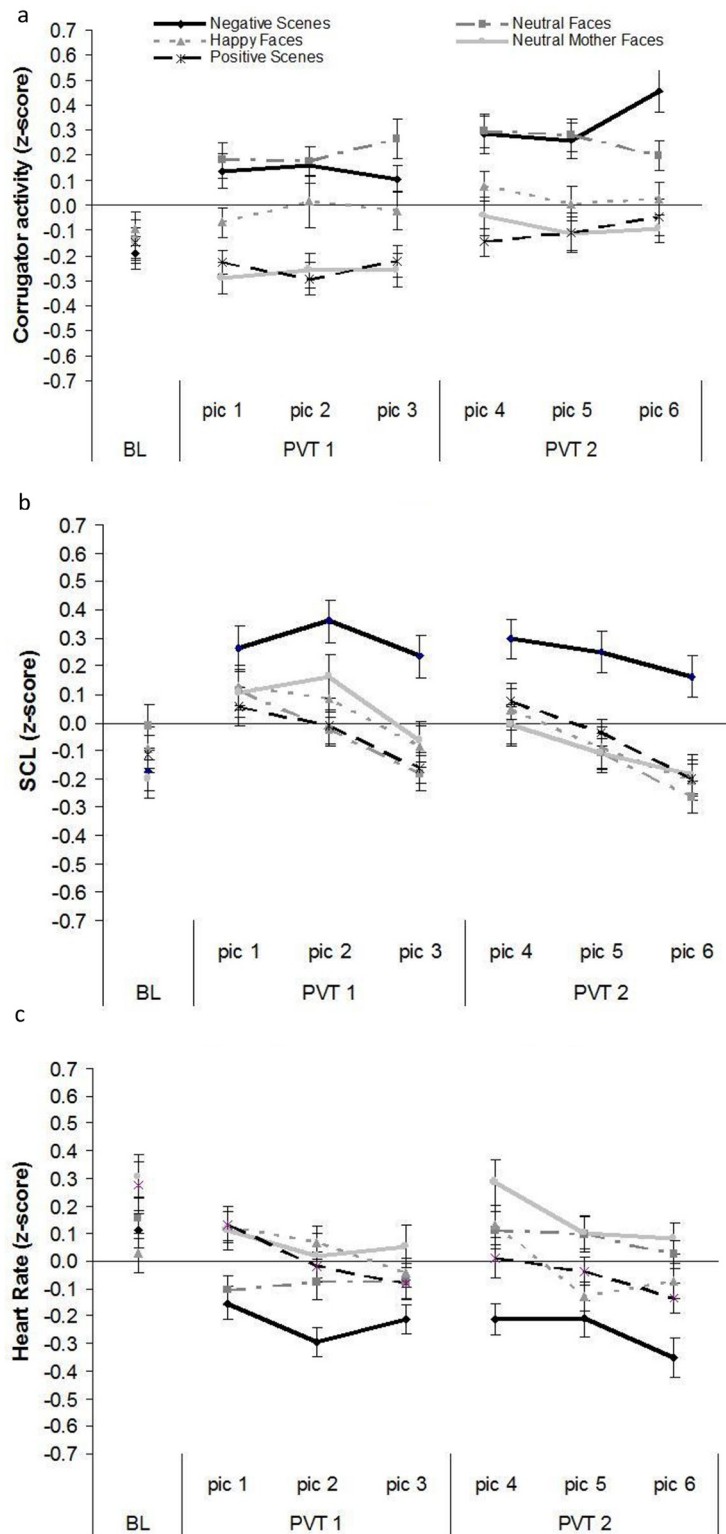

**Fig 5. Time course of psychophysiological responses.** Responses were intraindividually z-transformed and recorded during tonic heat pain stimulation combined with picture viewing. Shown are 5 s averages of (a) corrugator activity, (b) skin conductance levels (SCL), and (c) heart rate (HR) for baseline (BL), picture viewing time 1 (PVT1) including pictures 1–3 (pic1-3) and PVT2 including pictures 4–6 (pic4-6).

At t1, despite the significant main effect, there were no differences in HR between neutral and other pictures. Additional post hoc test between all picture categories also revealed not significant differences. At t2, there were no differences in HR between neutral and other pictures. At t3, HR was higher for neutral faces as compared to negative scenes ($d_{rm}$ = 1.09).

## Correlational analyses

**Pain threshold, experimental temperature, and pain self-report at baseline.** The mean pain threshold temperature, experimental temperature, and pain self-report at baseline ($M$ = 41.23°C, $SD$ = 3.29; $M$ = 42.1 °C, $SD$ = 2.94; $M$ = 8.04, $SD$ = 3.76; for more detail see also S2 Table) did not correlate significantly with any of the questionnaire reports ($|rs| < .15$, $ps > .356$).

**Affective pain modulation.** There were no significant correlations of posPM with measures of psychological health, pain catastrophizing, picture induced valence or arousal for both time points (t2, t3; see Table 4). Therefore, no partial correlations were calculated. More depressed (DTC) and more anxious (CAT-II) children rated positive scenes as significantly more arousing.

There were no significant correlations of negPM with measures of psychological health, pain catastrophizing, picture induced valence or arousal for both time points (t2, t3; see Table 4). Therefore, no partial correlations were calculated. Pictures rated as less positive (lower valence) had a stronger pain increasing effect at t3.

**Social pain modulation.** *Mothers' neutral faces.* Table 5 shows that more anxious children (CAT-II) and children with more behavioral difficulties (SDQ) experienced greater pain relief

**Table 4. Correlation of questionnaires with affective pain modulation and induced valence.**

| | PosPM t2 | PosPM t3 | NegPM t2 | NegPM t3 | Picture ratings | | | |
|---|---|---|---|---|---|---|---|---|
| | | | | | Positive scenes | | Negative scenes | |
| | | | | | valence | arousal | valence | arousal |
| | *r* | *r* | *r* | *r* | *r* | *r* | *r* | *r* |
| Child report: | | | | | | | | |
| Catastrophizing (PRCQ-C) | -.01 | .06 | .13 | -.02 | .04 | .26 | -.15 | -.23 |
| Depression (DTC) | .13 | -.04 | -.14 | .03 | .09 | **.34**[a] | -.12 | .06 |
| Anxiety (CAT-II) | .21 | -.00 | -.28 | -.05 | .08 | **.43**[b] | -.19 | .16 |
| Mother report: | | | | | | | | |
| Difficulties (SDQ–Problem score) | -.01 | .09 | .05 | .12 | .14 | .10 | -.17 | .11 |
| Picture Ratings: | | | | | | | | |
| Pos. scenes valence | .04 | .05 | n.a | n.a. | 1.00 | | | n.a. |
| Pos. scenes arousal | .21 | -.06 | n.a | n.a. | .05 | 1.00 | n.a. | |
| Neg. scenes valence | n.a. | n.a | -.17 | **-.34**[a] | .07 | n.a. | 1.00 | |
| Neg. scenes arousal | n.a. | n.a | -.21 | -.02 | n.a. | **.53**[c] | **-.38**[a] | 1.00 |

Pearson correlation coefficients of questionnaires with positive and negative affective pain modulation and induced valence and arousal by positive and negative scenes.

[a]$p < .05$

[b]$p < .01$

[c]$p < .001$

Note: Significant differences are highlighted in bold

Abbreviations: n.a. = not applicable; PRCQ-C = Pain Related Cognitions Questionnaire for Children; DTC = Depression Test for Children; CAT-II = Children Anxiety Test II; SDQ = Strength and Difficulties Questionnaire; posPM = positive pain modulation: refers to the difference when viewing neutral faces versus positive scenes, posPM > 0 indicates a pain relieving effect of positive scenes; negPM = refers to the difference when viewing negative scenes versus neutral faces, negPM > 0 indicates a pain enhancing effect of negative scenes.

**Table 5. Correlation of questionnaires with social pain modulation and induced valence.**

| | motherPM t2 | motherPM t3 | Picture ratings of mothers' faces | |
| --- | --- | --- | --- | --- |
| | | | valence | arousal |
| | *r* | *r* | *r* | *r* |
| <u>Child report:</u> | | | | |
| Catastrophizing (PRCQ-C) | .23 | **.38**[a] | **.31**[a] | **.39**[a] |
| Solicitous response (PPBI-C) | .11 | -.07 | -.04 | .01 |
| <u>Mother report:</u> | | | | |
| Catastrophizing about the child's pain (PCS-P) | -.15 | -.14 | -.01 | -.15 |
| <u>Child report:</u> | | | | |
| Depression (DTC) | .28 | .09 | .06 | .30 |
| Anxiety (CAT-II) | **.39**[a] | .11 | .09 | .30 |
| <u>Mother report:</u> | | | | |
| Difficulties (SDQ–Problem score) | **.34**[a] | .30[+] | .03 | .23 |
| <u>Child report:</u> | | | | |
| Mother support (CSI) | .02 | -.07 | -.24 | -.08 |
| Mother approval (CSI) | .18 | .03 | -.19 | .06 |
| <u>Picture Ratings:</u> | | | | |
| Mothers' faces valence | .21 | .16 | 1.00 | |
| Mothers' faces arousal | **.55**[c] | **.42**[b] | **.39**[a] | 1.00 |

Pearson correlation coefficients of questionnaires with social pain modulation by mothers' faces and induced valence and arousal by mothers' faces.

[a]$p < .05$

[b]$p < .01$

[c]$p < .001$.

Note: Significant correlations are highlighted in bold

Abbreviations: PRCQ-C = Pain Related Cognitions Questionnaire for Children; PPBI-C = Pain-related Parent Behavior Inventory–Child version; PCS-P = Pain Catastrophizing Scale–Parent version; DTC = Depression Test for Children; CAT-II = Children Anxiety Test II; SDQ = Strength and Difficulties Questionnaire; CSI = Childrearing Style Inventory; motherPM = mothers' faces pain modulation: refers to the difference when viewing neutral strangers' faces versus neutral mothers' faces, motherPM > 0 indicates a pain relieving effect of mothers' faces; t2 = after pictures 1–3, t3 = after pictures 4–6.

when viewing their mothers' faces and children who perceived their mothers' faces as more arousing experienced greater pain relief when watching their mothers' faces (t2). Children who had a stronger tendency to catastrophize about pain and who perceived their mothers' pictures as more arousing experienced greater pain relief while watching their mothers' faces (t3). Furthermore, children higher in pain catastrophizing rated their mothers' pictures to be more pleasant.

Due to the significant correlation between perceived arousal of mothers' faces and children's pain catastrophizing (t3; PRCQ-C; $r = 0.39$, $p = .011$), partial correlations were calculated (see Fig 6). They revealed that the association between children's pain-related catastrophizing and motherPM was no longer significant when controlled for arousal induced by mothers' faces ($r_{\text{PRCQ-C-motherPM.arousal}} = .25$, $p = .110$), yet, that the association between arousal and motherPM$_{t3}$ remained significant ($r_{\text{arousal-motherPM.PRCQ-C}} = .32$, $p = .042$) when controlled for pain-related catastrophizing.

*Strangers' happy faces*. More depressed (DTC) and more anxious (CAT-II) children showed greater pain relief when watching happy faces and children who rated happy faces to be more pleasant experienced greater pain relief (t2; Table 6). No partial correlations were calculated because no questionnaire data correlated significantly with both happyPM and valence or arousal ratings.

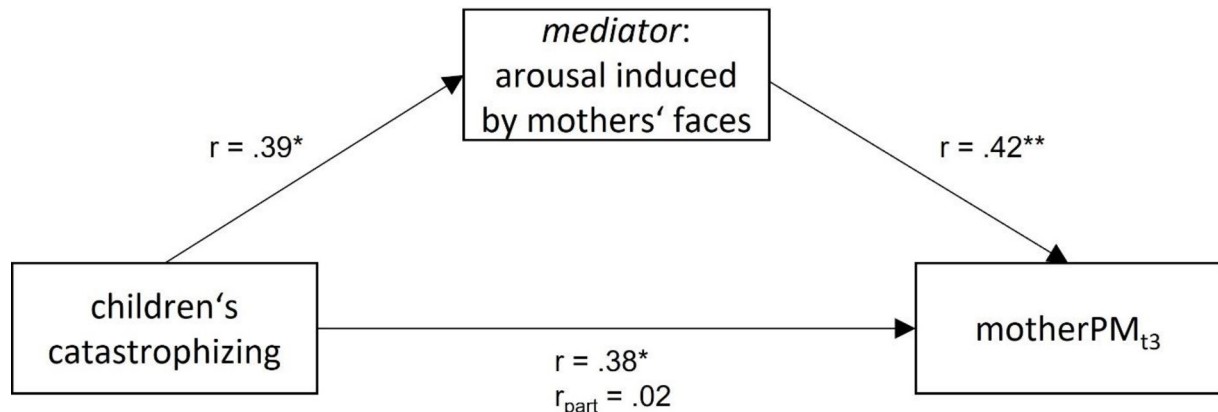

**Fig 6. The relation between children's catastrophizing and social pain modulation.** At t3 the relation between children's pain-related catastrophizing and social pain modulation is mediated by the arousal induced by mothers' faces. $*p < .05$, $**p < .01$, $r_{part}$: partial correlation of children's pain-related catastrophizing with motherPM$_{t3}$ when controlled for arousal induced by mothers' faces.

## Discussion

The main aim of the present study was to investigate in children the pain modulating effect of pictures varying in affective quality and social content. Pain modulation was evaluated both at the subjective and physiological level. Additionally, we examined variables such as parental pain behavior, children's pain-related catastrophizing, psychological health, and picture-induced affect for their possible impact on pain modulation.

Affective and social-affective pain modulating effects emerged both at the subjective and psychophysiological level. Reported pain intensity was reduced when viewing positive scenes (t2 and t3) and mothers' (t2 only) as compared to neutral faces. Further, corrugator activity was lower for all three positive categories, i.e. positive scenes, mothers' and happy faces, than

**Table 6. Correlations of pain modulation of happy faces with questionnaires and picture induced valence and arousal.**

|  | happyPM t2 | happyPM t3 | Picture ratings of happy faces | |
|---|---|---|---|---|
|  |  |  | valence | arousal |
|  | *r* | *r* | *r* | *r* |
| Child report: |  |  |  |  |
| Catastrophizing (PRCQ-C) | -.19 | .18 | .02 | .18 |
| Depression (DTC) | **.37**[a] | .13 | -.06 | .03 |
| Anxiety (CAT-II) | **.38**[a] | .01 | -.11 | -.02 |
| Mother report: |  |  |  |  |
| Difficulties (SDQ–Problem score) | .01 | -.02 | .15 | .01 |
| Picture Ratings: |  |  |  |  |
| Happy faces' valence | .10 | **.34**[a] | 1 |  |
| Happy faces' arousal | -.03 | .20 | .16 | 1 |

[a]$p < .05$

Note: Significant differences are highlighted in bold

Abbreviations: PRCQ-C = Pain Related Cognitions Questionnaire for Children; DTC = Depression Test for Children; CAT-II = Children Anxiety Test II;

SDQ = Strength and Difficulties Questionnaire; happyPM = pain modulation with happy faces: refers to the difference when viewing neutral faces versus happy faces, happyPM $> 0$ indicates a pain relieving effect of happy faces; t2 = after pictures 1–3, t3 = after pictures 4–6.

for neutral faces. Viewing happy faces and negative scenes had no significant effect on reported pain intensity, however. Negative affective pain modulation was found with regard to SCL and HR only, specifically viewing negative scenes resulted in higher SCL and lower HR (only PVT2) when compared to neutral faces.

The obtained *positive affective pain modulation* at the level of the subjective pain experience in our pediatric sample is consistent with studies in adults using IAPS pictures [9–12]. Importantly, the affective pain modulation was not limited to the subjective pain report, but also reduced pain-related facial (i.e. corrugator) acitivity. Aside from reflecting stimulus unpleasantness, reduced corrugator activity, i.e. less squeeze of the eyebrow, closely matched the pain relief at the level of the subjective report and suggests a diminished facial response to pain when viewing positive scenes. None of children's characteristics accounted for positive affective pain modulation. This may be because the questionnaire scores were all within normal ranges. Possibly, results would be different in samples with clinically elevated questionnaire scores [47].

Most interestingly, *social-affective pain modulation* was also demonstrated on the subjective level for mothers' faces, and regarding corrugator activity for mothers' and happy faces. The pain reduction by mothers' faces corresponds well with findings of adults showing pain relief when viewing pictures of significant others [13–16] and even extends some findings as we carefully controlled the mothers' facial expressions to be neutral, did not rely on personal pictures provided by participants, and did not instruct children to think of their mothers when looking at the mothers' photos. This pain-alleviating effect might be accounted for by mothers' faces inducing a positive affective state [48] or by the activation of feelings of being loved and supported [14]. Interestingly, social modulation by mothers' faces was affected by children's characteristics. More anxious children, children having more behavioral and emotional problems and children higher in catastrophizing benefitted more from viewing their mothers' faces. Intriguingly, the impact of catastrophizing on social pain modulation by the mothers' picture was mediated by perceived arousal of mothers' faces. We recently observed a similar mediation effect in women with perceived arousal of partners' pictures accounting for the relationship between pain catastrophizing and partner pain modulation [16]. Since catastrophizing has a social communicative function [1, 49], it is not surprising that we only observed its impact on social pain modulation. Since higher pain catastrophizing is related to lower self-regulation [50], children higher in catastrophizing may have a stronger tendency to turn to parents for support when attempting to regulate their pain. The mediation effect also corresponds well with assumptions of the communal coping model of pain catastrophizing [51], in which pain catastrophizing is conceptualizes as a coping response eliciting support and empathic responses from the social environment. This could account for the greater pain relief in high-catastrophizing children when viewing their mothers' pictures. The mediating effect of arousal is compatible with the notion that arousal indexes the degree of activation of a motivational system [52].

Results regarding social-affective pain modulation by happy faces are not as clear as those by mothers' faces. Children's valence and arousal ratings of happy faces were similar to those of mothers' faces, indicating that the affect induced by happy faces is similar to that induced by mothers' faces. Happy faces reduced corrugator activity, yet, did not alleviate pain intensity. This is inconsistent with our results in adults [16] where partners and happy strangers' faces alleviated perceived pain intensity. Other available picture studies did not measure pain-related facial activity [13–15]. With regard to the difference in pain relief between happy and mothers' faces, familiarity with the social interaction partner may be more important for social pain modulation in children than in adults. In adults, partners' and happy strangers' faces were found to serve as a social safety signal [16]. Children, however, may put more weight on

the unfamiliarity of strangers' faces than on their expressed affect, and, consequently, regard unfamiliar happy faces not as a safety signal. Interestingly, the modulating effect of happy faces on perceived pain intensity was also influenced by children's characteristics. More depressed, more anxious children and children having more behavioral and emotional difficulties not only benefitted from viewing their mothers' faces but also from viewing happy strangers' faces. Possibly, these children are more susceptible to positive signals from their social environment even when these come from strangers.

Findings regarding *negative affective pain modulation* are not as clear as those for positive affect. Negative scenes not only induced stronger arousal on a subjective (SAM ratings) and physiological (SCL) level, but also resulted in lower HR, consistent with the induction of negative affect. Higher SCL and lower HR had also been found in adult samples watching negative pictures [53–55]. Contrary to our expectations, however, negative affective scenes did not lead to an increase in reported pain intensity. Furthermore, neutral faces and negative scenes had similar corrugator activity despite differences in valence ratings. Possibly, negative scenes, despite being most unpleasant, caused affective states which in turn overrode the expected effects of negative affective pain modulation [56]. Since we did not assess induced mood, we can only speculate whether and which affective states were induced, e.g. positive excitement, comparable to the enjoyment of suspenseful films [57–60] or fear [61, 62] both of which could reduce pain sensations.

One main limitation of our study is the comparatively small sample size of children including a subsample of children suffering from recurrent pain. Notably, studies on affective pain modulation in adults [13–15] have comparable sample sizes. Further, we did not aim to compare healthy children with children with chronic pain, but collected information on pain to describe the participating children recruited in local schools. The number of participants suffering from recurrent pain was considerable, but did not allow to systematically compare both groups. Following a very conservative approach, we included this between-group variable in our analyses, yet, the lack of significant differences should not be generalized.

Another limitation is that we did not use a full factorial design including a happy expression of mothers, a neutral scene and possibly also negative emotional expressions of mothers and strangers, comparable to the negative scenes. We also had no "picture only" or "pain only" trial as we were concerned about the burden imposed on participating children when increasing the number of trials and, thereby, overall the length of the experiment which already was altogether 2.5–3.5 hours. Furthermore, it would have been difficult to obtain valid happy and negative mothers' facial expressions. A further limitation might be the use of pictures as they arguably have a limited ecological validity when compared to videos or actual presence of parents. Our picture viewing paradigm, however, ensures standardization and systematic variation of affect and social content. Lastly, due to limiting the burden on participants, we could only assess a small number of potential pain modulation determinants, thereby, omitting others such as mother-child relationship quality.

Beside these limitations, our study contributes to the understanding of affective and social-affective pain modulation in children. We could show that positive scenes and mothers' faces and to some extent happy strangers' faces had a pain-alliviating effect on children. Additionally, childrens's characteristics affected social-affective pain modulation. Particularly, more depressed, more anxious children and children who had more difficulties benefitted from viewing their mothers' and happy stranger faces. Further, the association between children's tendency to catastrophize and social-affective pain modulation was mediated by arousal induced by mothers' faces. Clinically, our results imply that simple measures such as pictures of mothers (and likely also fathers) or positive scenes could decrease children's pain experience whenever parents cannot be present (e.g. accidents occurring in kindergarden or school) or

when the actual presence of a parent may have negative effects on the children e.g. because parents have a high pain-related fear or engage in distressing behaviors.

## Supporting information

**S1 Table. Valence and arousal ratings of pictures from an internet search.** Ratings of 12 pictures of an internet search by 8 to 14-year-old children in school, class by class. Self-assessment manikins were used for valence and arousal ratings (SAM; [41]. Pictures were presented for 6 s each in different picture sets on a screen in front of the class.
(DOCX)

**S2 Table. Data of experimental pain measures and pain modulation.** Means and standard deviations (SD) for experimental pain measures, affective, and social pain modulation for the total sample as well as for healthy children and children with recurrent pain separately and the comparison between the two latter groups.
(DOCX)

## Acknowledgments

Authors wish to thank Kristina Braun, Simon Goff, Lea Lockmann and Cate Trillmich for their help with data collection, Laurens Berthold for his help with data analysis, and all participating children and their mothers for their support.

## Author Contributions

**Conceptualization:** Katrin Hillmer, Christiane Hermann.

**Data curation:** Katrin Hillmer.

**Formal analysis:** Katrin Hillmer.

**Investigation:** Katrin Hillmer.

**Methodology:** Christiane Hermann.

**Supervision:** Christiane Hermann.

**Writing – original draft:** Katrin Hillmer, Judith Kappesser.

**Writing – review & editing:** Judith Kappesser, Christiane Hermann.

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
