## [Decision Letter · Decision Letter 0]

9 Aug 2024

PONE-D-24-21539Affective and social pain modulation in children – experimental evidence using picture viewingPLOS ONE

Dear Dr. Kappesser,

Thank you for submitting your manuscript to PLOS ONE. After careful consideration, we feel that it has merit but does not fully meet PLOS ONE’s publication criteria as it currently stands. Therefore, we invite you to submit a revised version of the manuscript that addresses the points raised during the review process.

We look forward to receiving your revised manuscript.

Kind regards,

José A Hinojosa, Ph.D.

Academic Editor

PLOS ONE

3. In the online submission form, you indicated that [Insert text from online submission form here]. 

Additional Editor Comments (if provided):

Reviewers' comments:

Reviewer's Responses to Questions

**Comments to the Author**

1. Is the manuscript technically sound, and do the data support the conclusions?

Reviewer #1: Yes

2. Has the statistical analysis been performed appropriately and rigorously? 

Reviewer #1: Yes

3. Have the authors made all data underlying the findings in their manuscript fully available?

Reviewer #1: Yes

4. Is the manuscript presented in an intelligible fashion and written in standard English?

Reviewer #1: Yes

5. Review Comments to the Author

Reviewer #1: This is an interesting study that experimentally explores pain modulation (using positive, negative, and social stimuli) in children by analyzing self-reported and psychophysiological measures (i.e., SCL, HR, and corrugator EMG). However, my main concern is with the analysis, so I have some questions and suggestions that might help improve the manuscript:

1. Abstract (page 2, line 28). Please consider changing to “…extends previous findings in adults”.

2. Introduction (page 3). The introduction is clear and well structured, with references to a theoretical model and previous studies (mainly on adult samples), but is somewhat brief (e.g., no reference is made to the relevance of using psychophysiological measures to measure pain modulation). Previous studies have reported that subjective valence and arousal correspond to facial EMG and SCL, respectively (see https://doi.org/10.1016/j.biopsycho.2020.107974).

3. Introduction (page 4, lines 59-60). Please consider changing to “…modulating effect of pictures varying in affective quality (positive and negative scenes, happy and neutral faces) and social content (mother, strangers faces)…”.

4. Introduction (page 4, line 65). Please consider changing to “We hypothesized...”.

5. Materials and methods (page 4, lines 73-74). The data collection was carried out about 12 years ago. Why?

6. Materials and methods (page 5, lines 85-86). Much of the research indicates that despite altered patterns of interaction in adolescents, parent-child relationships remain important social and emotional resources well beyond the childhood years (see https://doi.org/10.1002/9780471726746.ch11). However, because the sample includes a wide age range (8-13 years), have possible differences due to age been explored?

7. Materials and methods (page 9, line 186). Was manual dominance assessed with a validated instrument?

8. Materials and methods (page 17, lines 367-369). Why did you use pain modulation indices only for correlational analyses?

9. Materials and methods (page 19, lines 416-418). This information is contradictory to the information in page 5, lines 89-91 ("15 of the 42 children reported heterogenous problems with recurrent pain [abdominal pain (n=13) headaches (n=7)] which is a considerable number, yet not large enough to allow for a systematic comparison between both groups").

10. Materials and methods (page 19, lines 418-421). Was the assumption of normality tested?

11. Materials and methods (page 20, lines 444-447). This information should be at the beginning of this subsection ("6.1. Analyses of variance").

12. Results (pages 23-24, lines 504-508). I do not understand why these results are reported in this sub-section when there is another sub-section dedicated to correlation results.

13. Results (page 28, lines 596-599). Reference to Table 5 is missing.

6. PLOS authors have the option to publish the peer review history of their article (what does this mean?). If published, this will include your full peer review and any attached files.

Reviewer #1: **Yes: **Carolina Sitges

---

## [Author Response · Author response to Decision Letter 0]

22 Sep 2024

We made changes to the manuscript and figures included so that they now meet PLOS ONE’s style requirements. 

2. We note that you have indicated that there are restrictions to data sharing for this study. 

We would like to point out that the data underlying the statistical data analysis as well as the research materials can in general be made available. The data which were collected in this experiment are not freely and directly available because the original approval by the local ethics committee of the Faculty of Psychology and Sports Science, Justus-Liebig University Giessen, Germany (#2010-0018) and the informed consent form signed by the participants did not include such direct access. However, the anonymized data will be made available to interested researchers upon request by the local ethics committee. For this purpose, its email address is given in the revised Data Availiability statement. This statement now reads:

Data Availability statement: 

The data which were collected in this experiment are not freely and directly available because the original approval by the local ethics committee of the Faculty of Psychology and Sports Science, Justus-Liebig University Giessen, Germany (#2010-0018) and the informed consent form signed by the participants did not include such direct access. However, the anonymized data will be made available to interested researchers upon request by the local ethics committee of the FB06 -Psychology and Sports Science. For this purpose, its contact email (ethikkommission@fb06.uni-giessen.de) is provided in the manuscript.

3. In the online submission form, you indicated that [Insert text from online submission form here]. All PLOS journals now require all data underlying the findings described in their manuscript to be freely available to other researchers, either 1. In a public repository, 2. Within the manuscript itself, or 3. Uploaded as supplementary information. This policy applies to all data except where public deposition would breach compliance with the protocol approved by your research ethics board. If your data cannot be made publicly available for ethical or legal reasons (e.g., public availability would compromise patient privacy), please explain your reasons on resubmission and your exemption request will be escalated for approval. 

As stated above, direct access of data had not been included either in the original approval by the ethics committee or in the informed consent form signed by participants. However, we can make the anonymized data available to interested researchers upon request to the local ethics committee.

We changed the information accordingly. The whole paragraph now reads:

For their participation children received 30 €. The study protocol had been approved by the local ethics committee (“Lokale Ethikkommission des Fachbereichs 06 Psychologie and Sportwissenschaften der Justus-Liebig Universität”, LEK-FB06; #2010-0018; https://www.uni-giessen.de/de/fbz/ fb06/psychologie/ethikkommission) stating “Decision: There are no ethical or profession law objections to the project”. Written informed consent was provided by all participating children and their parents.

We require you to either (1) present written permission from the copyright holder to publish these figures specifically under the CC BY 4.0 license, or (2) remove the figures from your submission.

The face stimuli were taken from the RaFD face stimuli set. On their homepage (https://rafd.socsci.ru.nl/FAQ.html) the authors answer the question “Am I allowed to use the RaFD faces in publications, e.g. journal articles or presentations about my research?” stating “Yes, in strictly scientific publications RaFD images can be presented as stimulus examples.” 

Similary, on their homepage (https://kdef.se/faq/using-and-publishing-kdef-and-akdef), the authors of the KDEF stimuli set require scientific users of their stimuli to include information on the KDEF image id in the figure text and to include a list of KDEF ids of all stimuli used in the experiment in the Method section. With regard to publishers, they state that “Researchers may always include sample images from KDEF in his/her manuscript when said manuscript is a doctoral thesis OR is a manuscript submitted to a scientific journal. A publisher may regard this mail as a written consent for such publication, or contact me (contact info below) directly if needed. For the KDEF stimuli, such a journal is typically PLOS ONE, EMOTION, NEUROPSYCHOLOGIA, COGNITION & EMOTION, SOCIAL COGNITIVE & AFFECTIVE NEUROSCIENCE, BIOLOGICAL PSYCHOLOGY, NEUROIMAGE, FRONTIERS IN PSYCHOLOGY, JOURNAL OF NEUROSCIENCE or PSYCHONEURO-ENDOCRINOLOGY. Or similar. … Apart from the above publication purposes, the KDEF and AKDEF stimuli may NOT be redistributed or shared without written consent from the copyright holder (Karolinska Institutet, Psychology section …).”

We looked at the IAPS website where we could not find any information regarding copyright. We contacted the authors personally but got no reply. Therefore, we removed the two IAPS picture examples from Figure 1. 

Comments Reviewer #1: 

1. Abstract (page 2, line 28). Please consider changing to “…extends previous findings in adults”.

Thank you for your suggestion! We changed the sentence accordingly. 

2. Introduction (page 3). The introduction is clear and well structured, with references to a theoretical model and previous studies (mainly on adult samples), but is somewhat brief (e.g., no reference is made to the relevance of using psychophysiological measures to measure pain modulation). Previous studies have reported that subjective valence and arousal correspond to facial EMG and SCL, respectively (see https://doi.org/10.1016/j.biopsycho.2020.107974).

Thank you for pointing this out! We realized how little information we provided on our use of psychophysiological measures. Therefore, we added a small paragraph including relevant information and references before the final paragraph of the Introduction. The last two paragraphs of the Introduction now read:

It is of interest whether pain modulation occurs not only on the level of subjective pain report but also on the level of psychophysiological measures. Pain as highly relevant stimulus induces autonomic arousal as indicated by higher skin conductance level (SCL) (26,27), heart rate (HR) (26,27), and higher activity of the corrugator muscle (assessed by electromyography; EMG), the latter one corresponding to eyebrow squeezing as core element of the facial pain expression (29) and correlating with perceived aversiveness of stimuli (30). A recent study with adults found effects of corrugator EMG, SCL and HR on social and affective pain modulation (REF). 

The present study aimed to investigate the pain modulating effect of pictures varying in affective quality (positive and negative scenes, happy and neutral faces) and social content (mothers’, strangers’ faces) in children during tonic heat pain stimuli. Pain modulation was assessed by self-report and physiological measures assessing autonomic arousal [(SCL, HR, and corrugator EMG; (18, 19)], and pain-related facial responses, more specifically corrugator muscle activity, a core element of the facial expression of pain (20) (21). We hypothesized that pictures of (a) negative scenes enhance, and positive scenes relieve pain, (b) neutral familiar (mother) and positive unfamiliar faces (happy stranger) alleviate pain. Finally, the influence of parental and child characteristics on children’s pain modulation was tested.

3. Introduction (page 4, lines 59-60). Please consider changing to “…modulating effect of pictures varying in affective quality (positive and negative scenes, happy and neutral faces) and social content (mother, strangers faces)…”.

Thank you again for this suggestion! We changed the sentence accordingly. It now reads: 

The present study aimed to investigate the pain modulating effect of pictures varying in affective quality (positive and negative scenes, happy and neutral faces) and social content (mothers’, strangers’ faces) in children during tonic heat pain stimuli.

4. Introduction (page 4, line 65). Please consider changing to “We hypothesized...”.

Thank you for your suggestion! We changed the sentence accordingly. 

5. Materials and methods (page 4, lines 73-74). The data collection was carried out about 12 years ago. Why?

The data collection was carried out by Katrin Hillmer (maiden name Damm; see ethics form) as part of her PhD. She started her training as clinical psychologist when she was about to finish data collection for this study. The clinical training in Germany takes about five years to complete without much time for other work such as research. Therefore, work on her PhD was delayed. When she finished her clinical training, she got married, moved away from Giessen and had two children all of which further delayed writing up her PhD work. 

6. Materials and methods (page 5, lines 85-86). Much of the research indicates that despite altered patterns of interaction in adolescents, parent-child relationships remain important social and emotional resources well beyond the childhood years (see https://doi.org/10.1002/9780471726746.ch11). However, because the sample includes a wide age range (8-13 years), have possible differences due to age been explored?

We agree with you that parent-child relationship remains an important social and emotional resource well beyond childhood years. And it might be true that there are age differences, yet, our sample is not large enough to allow for systematic comparisons taking child-parent relationship and changes depending on age into account. Further, it seems likely that also within one age group (e.g. younger or older children) there is considerable variability in quality of parent-child relationship. 

To account for your comment we included the following sentence in the limitations in the Discussion section:

Lastly, due to limiting the burden on participants, we could only assess a small number of potential pain modulation determinants, thereby, omitting others such as mother-child relationship quality.

7. Materials and methods (page 9, line 186). Was manual dominance assessed with a validated instrument?

Yes, manual dominance was assessed using the following five items of the Edinburgh Handedness Inventory (Oldfield, 1971; Williams, 2020): Writing, Drawing, Throwing, Scissors, Toothbrush. 

8. Materials and methods (page 17, lines 367-369). Why did you use pain modulation indices only for correlational analyses?

We followed a two-step approach. 

In a first step we exploited the repeated measures design to determine within-subject modulation in pain response depending on the direct comparison of picture categories, particularly affective vs. neutral and social-affective vs. neutral, and time course. A major advantage of this approach is that each participant serves as his/her own control, thus controlling for interindividual differences in the response level. This important information would be lost, if only pain modulation indices, i.e. a difference score, would have been considered.

In a second step, in order to determine the relationship between the degree of pain modulation and valence/arousal ratings and questionnaire data we had to rely on pain modulation indices since these allow an estimate of the magnitude of the modulation.

9. Materials and methods (page 19, lines 416-418). This information is contradictory to the information in page 5, lines 89-91 ("15 of the 42 children reported heterogenous problems with recurrent pain [abdominal pain (n=13) headaches (n=7)] which is a considerable number, yet not large enough to allow for a systematic comparison between both groups").

We are sorry for being seemingly contradictory! In the Material and Methods section we continue after the sentence you cite stating that 

Therefore, we decided to test for statistically significant differences between both groups and to only include children with recurrent pain if they do not differ systematically from healthy children in relevant outcomes. 

To make this idea clearer, we changed the sentence you cited by removing “yet not large enough to allow for a systematic comparison between both groups”, so that the relevant sentences now read: 

To be able to better describe our sample of children recruited in local schools, we included a question on recurrent pain. 15 of the 42 children reported heterogenous problems with recurrent pain [abdominal pain (n=13) headaches (n=7)] which is a considerable number. Therefore, in an attempt to control for a possible confounding influence of recurrent pain, we decided to check for statistically significant differences between both groups and to only include children with recurrent pain if they do not differ systematically from healthy children in relevant outcomes.

10. Materials and methods (page 19, lines 418-421). Was the assumption of normality tested?

Regression models, including repeated measures ANOVAs, were found to be robust to violations of normality when sphericity assumption is met (e.g. Blanca et al., 2017). Further, our sample size (N >= 30) is sufficiently large to forgo checking for normal distribution, since according to the central limit theorem (Bortz & Schuster, 2010) the sample distribution will be approximately normally distributed.

11. Materials and methods (page 20, lines 444-447). This information should be at the beginning of this subsection ("6.1. Analyses of variance").

We very much agree with you and changed the introductory paragraphs on “Analyses of variance” accordingly, so that they now read: 

We used the GLM procedure to conduct mixed design ANOVAs to analyze valence and arousal, pain intensity modulation, and psychophysiological correlates depending on picture category and time as within-subject factors. Since a number of children reported recurrent pain experiences we also included recurrent pain (“GROUP”) as between-subject factor to explore potential differences. Significant interactions with GROUP were followed by separate ANOVAS for healthy children and children with recurrent pain. Yet, since there were no significant effects for GROUP in any of the GLM analyses for the subjective and the physiological measures, only the results of the GLM analyses based on the total sample are reported. 

If the assumption of sphericity was violated, Greenhouse-Geisser corrected degrees of freedom (df) were used, yet, the nominal df are reported. For F-tests partial eta squared (η²) was calculated as effect size with 0.099 as threshold for small, 0.0588 for medium and 0.1379 for large effect sizes according to Cohen (43, 44). 

12. Results (pages 23-24, lines 504-508). I do not understand why these results are reported in this sub-section when there is another sub-section dedicated to correlation results.

We agree with you and moved this section to the results dedicated to correlations. This section now starts:

Correlational analyses

Pain threshold, expe

---

## [Decision Letter · Decision Letter 1]

29 Oct 2024

Affective and social pain modulation in children – experimental evidence using picture viewing

PONE-D-24-21539R1

Dear Dr. Kappesser,

We’re pleased to inform you that your manuscript has been judged scientifically suitable for publication and will be formally accepted for publication once it meets all outstanding technical requirements.

Kind regards,

José A Hinojosa, Ph.D.

Academic Editor

PLOS ONE

Additional Editor Comments (optional):

Reviewers' comments:

Reviewer's Responses to Questions

**Comments to the Author**

1. If the authors have adequately addressed your comments raised in a previous round of review and you feel that this manuscript is now acceptable for publication, you may indicate that here to bypass the “Comments to the Author” section, enter your conflict of interest statement in the “Confidential to Editor” section, and submit your "Accept" recommendation.

Reviewer #1: All comments have been addressed

2. Is the manuscript technically sound, and do the data support the conclusions?

Reviewer #1: Yes

3. Has the statistical analysis been performed appropriately and rigorously? 

Reviewer #1: Yes

4. Have the authors made all data underlying the findings in their manuscript fully available?

Reviewer #1: Yes

5. Is the manuscript presented in an intelligible fashion and written in standard English?

Reviewer #1: Yes

6. Review Comments to the Author

Reviewer #1: I consider that the authors have sufficiently addressed the issues raised during the review process and have improved the manuscript, so I consider it valid for publication.

7. PLOS authors have the option to publish the peer review history of their article (what does this mean?). If published, this will include your full peer review and any attached files.

Reviewer #1: **Yes: **Carolina Sitges

---

## [Editor Report · Acceptance letter]

20 Nov 2024

PONE-D-24-21539R1 

PLOS ONE

Dear Dr. Hermann, 

I'm pleased to inform you that your manuscript has been deemed suitable for publication in PLOS ONE. Congratulations! Your manuscript is now being handed over to our production team.

Kind regards, 

on behalf of

Dr. José A Hinojosa 

Academic Editor

PLOS ONE